# OpenReview forum: "PaCE: Parsimonious Concept Engineering for Large Language Models"
_NeurIPS.cc/2024/Conference — NeurIPS 2024 poster_

### Official Review · Reviewer_rWAy · 2024-06-30

**Soundness:** 4
**Presentation:** 4
**Contribution:** 3
**Rating:** 7
**Confidence:** 4

**Summary:**

This paper introduces Parsimonious Concept Engineering (PaCE), a novel framework for aligning LLMs by modifying their activation space. This framework constructs a large-scale concept dictionary in the activation space and partitions concepts into benign or undesirable categories. During inference, PaCE decomposes activations into these concept directions, removing undesirable components to reorient the LLMs' behavior towards alignment goals while preserving linguistic capabilities. The authors demonstrate the effectiveness of PaCE in tasks such as response detoxification, faithfulness enhancement, and sentiment revision.

**Strengths:**

- This paper proposes a novel activation manipulation framework, different from the existing methods of vector addition and orthogonal projection.
- The authors comprehensively and clearly articulate the deficiencies of existing methods and the ways in which the new framework addresses these deficiencies.
- The authors validate the superiority of the proposed method in pursuing alignment goals and preserving linguistic capabilities compared to existing methods across multiple tasks.

**Weaknesses:**

Overall, I did not find significant flaws that outweigh the contributions of this paper. The main weakness lies in the efficiency of the framework.

During inference, using this method results in each token's prediction taking approximately 2-3 times longer than when not using it. In contrast, simply using vector addition only increases the time by about 30-40%. Additionally, although the concept vocabulary only needs to be obtained once, the concept direction dictionary needs to be constructed and stored for different LLMs.

**Questions:**

* Could the authors provide more discussion on the computation time required and storage space (including memory usage and disk space) needed for constructing the direction dictionary?

* Why does OrthoProj require significantly more computation time in Table 2 compared to other methods? This seems to conflict with the statement in Remark 3 that OrthoProj is a special case of the method proposed in this paper.

**Limitations:**

Yes

---

> ### Author Rebuttal · Authors · 2024-08-06
>
> Dear Reviewer rWAy,
>
> Thank you for your insightful reviews and kind support for acceptance. It is our pleasure to reply to your comments.
>
> **Time efficiency**
>
> Thank you for discussing PaCE's time efficiency. We would like to emphasize that the main contributions of the paper are (1) a new dataset of concept dictionary and (2) a sparse coding process using this dictionary to decompose latent LLM representations. VecAdd and OrthoProj do not sufficiently model the representation space, and PaCE adopts a dictionary to have enhanced alignment performance.
> * PaCE is faster than OrthoProj  ([18, 79], Table 2) and we provide an option of a time-efficient solver (OMP, see discussion below).
> * PaCE has much higher averaged safety (Table 1), faithfulness and sentiment revising scores (Table 4) than VecAdd, OrthoProj, Prompting, and the vanilla model.
>
> Hence, the goal of this paper is to offer a competitively novel solution to representation manipulation for alignments of LLMs. We gladly take your comment and explore alternative optimization techniques for Equation 1 in the paper from the sparse coding literature that significantly reduces inference times. Below we briefly describe various solvers.
>
> * The elastic-net solver (Line 213) optimizes Equation 1 to obtain coefficients simultaneously.
> * We also consider Orthogonal Matching Pursuit (OMP), a fast greedy solver [D1, D2]. OMP iteratively adds concepts to the support based on maximum coherence with the residual and updates the residual until the support size k is reached. In other words, k is the number of non-zero coefficients.
>
> Comparing solvers on response detoxification (\S 4.1) shows certain setups of OMP are faster than Elastic Net (e.g., OMP with $k=50$ is an order of magnitude faster with 12.3% lower safety score). This demonstrates that a greedy solver improves speed at a performance cost. We will include the experiment in $\S4.1$ of the paper based on your suggestions.
>
> || OMP (k=50) | OMP (k=100) | OMP (k=150) | OMP (k=20) | Elastic Net |
> |---|---|---|---|---|---|
> | Time per decomposition (s) | 0.045 | 0.182 | 0.381 | 0.749 | 0.411 |
> | Safety (↑) | 63.1 | 64.4 | 66.9 | 70.8 | 72.0 |
>
> **Computing concept dictionary for different LLMs**
>
> Thank you for bringing up the point. We would like to explain our **motivation** for collecting the dataset, and how the cost may not be a concern in terms of our efforts in **open-sourcing** and **low amortized cost**.
>
> * (Motivation) We noted that the scale of existing datasets on the concept stimulus (e.g., RepE [79], ActAdd [64] in the paper) was limited. This incentivized us to (1) collect a large stimulus dataset to better model the concepts and (2) pre-compute the concept dictionary in an LLM representation space.
>
> * (Open-sourcing) Future researchers need not rerun dataset collection as we will open-source our dataset of human-readable concepts, stimuli, and concept dictionaries for LLaMA2 7B/13B and the Mistral 7B. Pre-computed concept partitions for downstream alignment tasks will also be shared."
>
> * (Low amortized cost) Each target model's concept dictionary needs to be extracted only once, with costs amortized across tasks. For example, extracting our dictionary for LlaMA2-7B took 25 minutes on an NVIDIA A40 machine and the dictionary was used for all evaluations on that model.
>
> **Specific time and space requirements for constructing the dictionary**
>
> We provide the statistics of computation and storage space needed for constructing the dictionary. We will include them as a new subsection (Appendix C.4) in our paper.
>
> * GPU memory usage for hosting the dictionary during each decomposition:
>   * LLaMA2 7B: 0.31 GB
>   * LLaMA2 13B: 0.45 GB
> * Local disk storage usage for storing dictionaries (for all involved 19 layers):
>   * LLaMA2 7B: 5.80 GB
>   * LLaMA2 13B: 8.26 GB
> * Time taken to compute and extract the dictionaries (NVIDIA A40 GPU machine):
>   * LLaMA2 7B: 25.2 minutes
>   * LLaMA2 13B: 40.8 minutes
>
> **Computation time of OrthoProj**
>
> * We appreciate your excellent comments on this point. The difference lies in the fact that the OrthoProj method computes a projection at multiple layers of the LLM to maintain a fair estimation of the coefficient. While PaCE uses a concept dictionary to model the representation space, OrthoProj does not sufficiently model it and could remove the contribution of desirable concepts. To maintain a stable steering process, the coefficient needs to be re-estimated at every decoder layer. Instead, VecAdd pre-sets a coefficient as a hyper-parameter and reuses it across the rest of the layers. Similarly, in experiments ($\S 4$ of the paper), PaCE performs decomposition at the first encountered intermediate layer and reuses it across the rest of the layers. In practice, one can freely tune the VecAdd coefficient through grid search and choose the layers where PaCE decomposition happens.
> * We find the choice of PaCE is sufficient for steering the LLM. We hypothesize this is because sparse coding can find a more suitable steering direction that directly reflects the desired concept change as compared to simply running orthogonal projection. Lastly, thank you for carefully reading the remarks. Your time and effort are greatly appreciated.
>
> Also, our paper’s Appendix B.4 includes the implementation details of frameworks and hyper-parameters for inferring on open-source LLMs. Based on your suggestion, we will include the justification in a new subsection Appendix B.5 and update the existing description to better clarify the difference between OrthoProj and other methods.
>
> Hopefully this alleviates your concerns; if not, please engage with us during the discussion phase.
>
> Thanks and regards,
>
> Authors of Submission #8804
>
> [D1] _Orthogonal matching pursuit: Recursive function approximation with applications to wavelet decomposition._ Pati et al., 1993.
>
> [D2] _Orthogonal matching pursuit for sparse signal recovery with noise._ Cai et al., 2011.

---

> > ### Comment · Reviewer_rWAy · 2024-08-12
> >
> > Thank you for your effort in providing numerous complementary results. These address my concerns about efficiency and could greatly improve the soundness of the paper. As a result, I will maintain my current score in favor of acceptance.

---

> > > ### Author Response · Authors · 2024-08-13
> > > **Thank you for your reply and support!**
> > >
> > > Dear Reviewer,
> > >
> > > Thank you for your timely feedback and kind recognition of our evaluations. Your support of acceptance is very encouraging to us, and your insights have greatly improved our paper. We will take your suggestions on revising the paper as promised: the manuscript $\S 4$ and Appendix B will be revised to include our discussions on framework efficiency.
> > >
> > > Sincerely,
> > >
> > > Authors

---

### Official Review · Reviewer_75Pi · 2024-07-07

**Soundness:** 3
**Presentation:** 3
**Contribution:** 2
**Rating:** 3
**Confidence:** 5

**Summary:**

This paper introduces an activation engineering framework for LLM alignment. PaCE addresses the challenge of reducing undesirable outputs while retaining benign concepts through a two-stage process: Concept Construction and Partition, and Activation Decomposition and Intervention. It constructs a large-scale concept dictionary and uses sparse coding to decompose LLM activations and selectively remove undesirable components. The authors demonstrate PaCE's effectiveness on tasks like response detoxification, faithfulness enhancement, and sentiment revising, showing improved alignment performance without compromising linguistic capabilities.

**Strengths:**

1. This paper presents a refined activation engineering method for alignment, achieving outstanding results especially with the 13B model.

2. The method is provided with a clear visualization and detailed interpretation to illustrate its underlying principles.

**Weaknesses:**

1. Quality of Concept Dictionary: the method's performance could be overly dependent on the quality and coverage of the concept dictionary. What measures are taken to ensure its comprehensiveness and that the dictionary is free from biases?

2. Static Activation: Static concept dictionary provides static activation while being decomposed, making PaCE unable to handle context-dependent concepts.

3. Without consideration of polysemy:  Many common concepts have different meanings in different scenarios. For example, the concept ‘kill’ is harmful when referring to the act of terminating a life, while it is benign in the usage of ‘killing time’. Both two usages are common. However, this paper ignores the polysemy and treats all 30 pieces of contextual stimuli same in constructing concept dictionary construction. Thus, I am a bit concerned about the accuracy of the extracted concept direction. Besides, the design of retrieval promotes the diversity of the concept stimuli, while may undoubtedly increases the potential of the situation that the same word in different or even opposite meanings.

4. Misuse of representation reading algorithm (arXiv:2310.01405) in extracting concept directions: In Appendix B.2 Eq.6, authors take a difference between the activations of concept t_i and all other concepts t_j (j≠i). However, Zou et al. (in Sec. 3.1) state that stimuli in the pair should have different target concept or function. Take detoxification task as an example, the concept ’bias’ and ‘narrowed’ in Fig. 7 are both harmful content which needed to be removed. Since ‘bias’ and ‘narrowed’ have the same target function, representation reading algorithm will not pair them to take a difference, while this paper also pairs them. This is quite weird.

5.  At the risk of compromising helpfulness: Following W3. and W4., the proposed concept directions may not be accurate and removing them may remove some benign concepts, thus limiting the model helpfulness.

6. Slower inference speed and time-efficiency: Since authors have claimed that ‘PaCE is computationally slower than VecAdd’, PaCE will undoubtedly slow down the original model’s inference speed, which is very important during model deployment.

**Questions:**

1. Can PaCE handle concepts that have multiple semantic meanings or are context-dependent?

2. Are the VecAdd/OrthoProj baselines constructed based on the same concept dictionary with PaCE during the experiments?

3. How to choose the intervention decoder layers: Prior works (arXiv:2312.06681) find the optimal layer for steering by sweeping over all layers and performing intervention to evaluate the effect. Out of curiosity, how do authors choose the layers?

4. Can authors provide an evaluation on XSTest dataset (arXiv:2308.01263) when applying PaCE in detoxification task? I am a bit concerned about that PaCE will increase the false refusal rate on benign queries.

5. The MMLU results about VecAdd in Tab. 1 are questionable according to (arXiv:2402.19465). I doubt if the choice of steering layers for VecAdd is also the same as the setting of PaCE (last 19 layers) which results in the significant performance drop in MMLU since for VecAdd choosing the middle layers is better.

**Limitations:**

In terms of experimental backbone models, it is recommended that the authors include more models (e.g., Mistral) not merely Llama2 Family for a comprehensive validation on the effectiveness of PaCE.

In terms of linguistic capability evaluation, only fluency, perplexity and one multi-choice QA benchmark is limited. Generation tasks like open domain QA (e.g., MT-Bench), summarization (e.g., XSum) are supposed to be evaluated.

---

> ### Author Rebuttal · Authors · 2024-08-07
>
> Thank you for your time on the paper. We are happy to address your comments below.
>
> **Quality of the concept dictionary**
>
> Excellent question! The efforts are made in two-fold: **the concept words** (indexes) and **the context stimuli** (contents) that define the semantics of each word.
>
> * For the **concept indexes**, we took the most frequent 40,000 words from the Brown Corpus ([14] in our paper), which models standard American English [C2] to ensure that our dictionary covers comprehensive concepts.
> * We defined the **meaning** of each concept via retrieving external knowledge from Wikipedia and instructing GPT4 to generate context stimuli. The semantics of a concept may develop over time: e.g., “mouse” was not regarded as a digital input device before the era of PC [C1]. We consider that GPT’s parametric knowledge is fixed at a timestamp. We propose to retrieve external knowledge for stimulus synthesis because the external knowledge is more updatable and explicit, which serves as enriched descriptions for concepts and helps GPT to generate more grounded and diverse scenarios. To further add to the transparency, we will open-source the dataset so that the community can reuse and improve on it.
>
> **Words have polysemy**
>
> We agree with you that concepts may have multiple meanings. We tried to be frank on this in our paper on Lines 954-956: “Current practice usually finalizes a single vector per concept by SVD, but theories on polysemanticity and recent studies on the causal models of language suggest that a concept might be better represented by a union of subspaces, each corresponding to different semantic meanings”.
>
> However, we argue that, while VecAdd and OrthoProj may be affected by polysemy, PaCE's overcomplete dictionary allows accurate analysis of the target representation through sparse decomposition. Tables 1 and 2 in the paper show PaCE outperforming OrthoProj and VecAdd on linguistic metrics. We further provide **empirical evidence** on MT-Bench (QA) and XSum (summarization) with **explanation** for polysemy not harming PaCE’s model helpfulness.
>
> **Empirical evidence.** Table 1 and Table 4 in $\S 4$ show that PaCE is better than OrthoProj and VecAdd on linguistic capability metrics (fluency, perplexity, MMLU). Based on your suggestions, we conduct further experiments on MT-Bench and XSum, whose implementation is detailed in the one-page PDF. PaCE maintains superior linguistic capability compared to VecAdd and OrthoProj.
>
> |MT-Bench Results||Vanilla|PE|VecAdd|OrthoProj| PaCE |
> |:---:|:---:|:---:|:---:|:---:|:---:|:---:|
> | LlaMA2-7B-Chat | GPT-4o as Judge (%, ↑) | 6.81 | 6.80 | 6.68 | 6.42 | 6.75 |
> | LlaMA2-13B-Chat | GPT-4o as Judge (%, ↑) | 7.02 | 6.95 | 6.83 | 6.94 | 6.96 |
>
> |XSum Results||Vanilla|PE|VecAdd|OrthoProj|PaCE|
> |:---:|:---:|:---:|:---:|:---:|:---:|:---:|
> | LlaMA2-7B-Chat | ROUGE-1 (%, ↑) |14.1| 14.6 | 13.7 | 14.4 | 14.6 |
> || BERT F1 (%, ↑) |82.2| 82.6 | 81.9 | 82.5 | 83.0 |
> || METEOR (%, ↑) |23.0| 22.9 | 22.0 | 23.1 | 22.5 |
> |LlaMA2-13B-Chat| ROUGE-1 (%, ↑) | 15.9 | 15.3 | 15.7  | 14.8 | 15.8 |
> || BERT F1 (%, ↑) |85.3 | 84.8  |84.7|84.9|85.2|
> || METEOR (%, ↑) |24.1 | 22.9  | 24.0|23.8|24.0|
>
> **Explanations**. We attribute this to the large-scale dictionary with sparse coding, explained as follows.
> * Since the dictionary is large, concepts with single and clear semantics are involved. E.g., if stimuli of “kill” may have different meanings, there exist other more polarized concept vectors such as “murder” (more harmful) and “spend” (more benign).
> * Sparse coding aims to choose the fewest concepts to reconstruct the latent representation (i.e., parsimony). For the sake of argument, assuming the sentence is about “killing time” and the vanilla LLM has the correct semantic understanding of its benignness, the latent representation of the whole sentence will be closer to concepts such as “spend” and “time” rather than string-matching to “kill” (which in your setup could have mixed harmful and benign senses). As the sparse coding of the target representation promotes the parsimonious selection of concepts with monosemantics, it helps to represent benign contexts correctly without assigning significant weights to ambiguous terms like "kill".
>
> **Response for pairing in representation reading; time efficiency; context-dependent concepts**
>
> Thank you for your insightful discussion. Due to OpenReview's limitations on figures, please refer to the one-page PDF under Global Response for our detailed responses.
>
> **Compliance to benign requests in the XSTest dataset**
>
> Thanks for inquiring on this very insightful dataset. PaCE decomposes input based on underlying semantics, not string-matching. It removes only malicious parts of the representation, preserving compliance with benign requests. We evaluate benign requests in XSTest, following the XSTest classification: I: full compliance, II: full refusal, III: partial refusal. Compliance is minimally affected. More detailed evaluations are in the one-page PDF.
>
> ||Vanilla|||PaCE|||VecAdd||
> |:---:|:---:|:---:|:---:|:---:|:---:|:---:|:---:|:---:|
> |I|II|III|I|II|III|I|II|III|
> |211|30|9|206|33|11|200|39|11|
>
> **Middle layers for representation manipulation**
>
> Thank you for this comment that allows us to correct a typo: In Line 909 (Appendix B.4), we wrote “activation vectors are extracted from the last 19 layers of the target LLM’s decoder layer.” We meant to say “activation vectors are extracted from the last-29th to the last-11th layer (totaling 19 layers) of the target LLM’s decoder layers”. Thus, the VecAdd in our experiments did use the middle layers.
>
> **PaCE on Mistral-7B**
>
> Please refer to our response and evaluation in **Generalization to Other LLMs** (Reviewer Cdr5).
>
> Thanks and regards,
>
> Authors of Submission #8804
>
> [C1] Biases in Large Language Models: Origins, Inventory, and Discussion. Navigli et al., 2023.
>
> [C2] Role of the Brown Corpus in the History of Corpus Linguistics. Kholkovskaia et al., 2017.

---

> > ### Author Response · Authors · 2024-08-13
> >
> > Dear Reviewer,
> >
> > We are grateful for your efforts and insights in the review, and we hope our response together with experiments covered your concerns. As the discussion phase is about to close, we are looking forward to hearing from you about any further feedback. We are more than happy to clarify or address additional comments (if any).
> >
> > Thanks and regards,
> >
> > Authors

---

### Official Review · Reviewer_s54k · 2024-07-08

**Soundness:** 3
**Presentation:** 2
**Contribution:** 3
**Rating:** 6
**Confidence:** 4

**Summary:**

This work proposes an activation engineering framework for alignment, i.e., PaCE, consisting of a task-driven concept dictionary and linear decomposition algorithm. Thus, the toxic or harmful input can be decomposed into a new activation vector combination, that removes the harmful activation. This is a generalization and adaptive method compared to VecAdd and OrthoProj. However, the presentation is somewhat confusing. It is expected that the authors can clarify it and improve the paper's readability.

**Strengths:**

This paper presents a novel safety alignment method based on activation engineering.

The evaluation of PaCE is comprehensive. The results are insightful.

**Weaknesses:**

There are some confusing parts (see detailed comments below). The presentation needs significant improvements.

There is a large time and computation cost in the preparatory phase. The efficiency is questionable.

It is necessary to analyze more serious jailbreaking examples, e.g., toxic，biased.

**Questions:**

It is not clear how the dictionaries and scores are provided to the LLMs along with the input? Why does this linear degradation performed at each layer of the LLM not cause a decrease in fluency and efficiency?

Is it possible to classify activation alignment as a scheme for the inhibition of malicious neurons or prompt purification? Compared to knowledge editing (e.g., "Detoxifying Large Language Models via Knowledge Editing"), what advantage of PaCE? Is the decomposition of semantic concepts the key to PaCE not affecting the normal response?

According to Remark 3 of 3.3, I understand that the limitation that exist in VecAdd and OrthoProj are directional deletion and some limitations claimed in Figure 2. So, is dictionary construction the key to helping PaCE address limitations such as ambiguous deletions? How is it that independence between concepts can be solved by sparsity?

Compared to RLHF, SFT, and other secure alignment schemes, I understand that the dataset of RLHF is replaced with a knowledge-driven concept dictionary, and the dynamic scoring during training is replaced with a task-driven concept splitter of GPT-4. If the understanding is correct, can the authors further explain the advantages?

As far as I know, the Chat model of the LLaMA series has strong security alignment capabilities, but the jailbreak examples given by the author are relatively weak. If possible, can the authors give a comparison of the detoxification of the more offensive or malicious jailbreak examples on AdvBench, for example?

**Limitations:**

The authors discussed limitations in Appendix B.6.

---

> ### Author Rebuttal · Authors · 2024-08-07
>
> ### Response to Reviewer s54k (1/2)
>
> Dear Reviewer s54k,
>
> Thank you for the thoughtful feedback and constructive questions. We address the concerns and provide additional evaluations to validate our work. We add the suggested improvements to enhance the quality of our work.
>
>
>
> **Preparatory Phase Concerns**
>
> Thank you for recognizing our efforts in dataset collection. Early in developing PaCE, we noted that the scale of existing datasets on the concept stimulus (e.g., RepE [79], ActAdd [64]) was limited. This incentivized us to (1) collect a large stimulus dataset to better model the concepts (2) pre-compute the concept dictionary in a larger LLM's representation space. Below we try to address your comments on computational costs in this phase.
>
> Future researchers need not rerun dataset collection as we will open-source our large dataset of human-readable concepts and stimuli. Further, each target model needs a concept dictionary to be extracted only once and this cost is amortized across different tasks on this model. For example, extracting our dictionary for LlaMA2-7B model took 25 minutes on our eight-card GPU machine and was used for all evaluations on that model. Lastly, despite this one-time preparation cost, the decomposition and intervention stages of PaCE are flexible in time efficiency and task performance. In Table 2 of the paper, we show that PaCE is faster than OrthoProj with a higher performance.
>
>
>
> **Providing dictionary and scores to LLM**
>
> We appreciate your question which helps improve the clarity of the paper! In short, dictionaries are a collection of concept vectors and are frozen for representation decomposition.
>
>
>
> * First, the LLM takes an input prompt (e.g., malicious requests)
> * Compared to a regular LLM, the representation engineering framework [B1] extracts the activations at each decoder block of the transformer. Such extraction results in a vector corresponding to the input prompt, which can then be modified for steering in different ways.
> * For PaCE, this steering has two main stages:
> * Stage 1 pre-computes the large-scale concept dictionary offline and the partition (i.e., scores) of which concepts represent benign/harmful concepts.
> * In Stage 2, at inference time, our approach extracts the representation of an input prompt and uses sparse coding to decompose this as the linear combination of atoms in our frozen dictionary. We then modify this linear combination by removing undesirable components and proceeding with inference in the LLM with the detoxified representation. We hope this clears up any misunderstandings.
>
> We will include the justifications in Appendix B.3 of our paper and revise other parts of the manuscript correspondingly.
>
>
>
> **Effects on fluency and efficiency, and the role of decomposition to preserve the normal response**
>
> Thank you for acknowledging the advantages of PaCE. Below we explain in two parts:  fluency (Part 1) and efficiency (part 2).
>
> Part 1: The key to keeping fluency is to maintain the benign components of the activation of user input prompts. PaCE handles it by decomposing the user input’s activation along benign and harmful concepts, while VecAdd and OrthoProj do not sufficiently model the concepts in the representation space. The decomposed solution of PaCE accurately estimates the benign components and preserves them during the intervention process.
>
> Part 2: The decomposition is efficient because the optimization problem for Equation (1) in the paper is convex with known fast solvers such as active-set (used in this paper) and OMP (see the discussion with Reviewer Cdr5/rWAy,).
>
>
>
> **Connection between activation alignment and the inhibition of malicious neurons or prompt purification**
>
> Thank you for bridging PaCE with other alignment schemes. If the concept vectors in PaCE are sparse/axis-aligned, i.e. each concept vector only has a few specific non-zero elements, then yes – subtracting such a vector from the activation is inhibiting the support neurons. We observe that concept vectors are dense and not axis-aligned, which means that many neurons are involved.
>
>
>
> **Detoxification on AdvBench**
>
> AdvBench uses GCG [B2] to adversarially optimize a jailbreak suffix for a harmful behavior request. In the table below, we show the LlaMA-7B-Chat safety score (%, ↑) on the effective set of suffix attacks for AdvBench harmful behavior set. We observe that PaCE outperforms other baselines. We also note that the outperformance of PaCE in our paper’s jailbreaks is more significant than that in suffix attacks. This could be because story-telling and roleplay jailbreaks (used in our paper) contain more complex and entangled concepts. Under this scenario, PaCE decomposes the target representation and well estimates the malicious component, while VecAdd and OrthoProj do not model the space sufficiently. In the AdvBench case, instead, the optimized adversarial suffix can be regarded as the text-space inversion of straightforward malicious concepts. PaCE and other defense mechanisms in latent space and prompt space shall effectively defend these suffixes more easily. We will include the evaluation in $\S 4.1$ of our paper.
>
>
> |   | Vanilla  | PE  | VecAdd  | OrthoProj  | PaCE (Ours) |
> |---|---|---|---|---|---|
> | LlaMA2-7B-Chat  | 11.72  | 91.90  | 94.51  | 92.81  | 96.65  |
> | LlaMA2-13B-Chat  | 18.04  | 93.86  | 95.33  | 96.72  | 99.17  |
>
>
>  (continued in the comment below)

---

> ### Author Response · Authors · 2024-08-07
>
> (rebuttal continued here)
> ### Response to Reviewer s54k (2/2)
>
>
> **Dictionary construction to address limitations such as ambiguous deletions (Part 1); Sparsity to handle independence between concepts (Part 2).**
>
> Part 1: Yes, PaCE has a large-scale dictionary modelling sufficient concept directions in the latent space, which allows PaCE to accurately analyze the compositionality of benign concept directions in the target.
>
> Part 2: When the concept vectors are not independent, infinitely many linear combinations of concept vectors can reconstruct the activation equally well.  Therefore, one needs regularization to break ties.
>
>
>
> * Sparsity serves as one regularization, which models the belief that the activation can be written as a linear combination of _only a few_ concept vectors. This is favorable since shorter explanations are deemed more interpretable [B3, B4].
> * A standard regularizer is the sum-of-squares of the combination coefficients, as in ridge regression. However, this tends to give combinations that use most of the concept vectors in the dictionary,
> * and since the dictionary is large this is not efficient.
>
>
>
>
> **Advantages of PaCE over RLHF, SFT, and Knowledge Editing (KE)**
>
> Your analogy of the correspondence among different alignment paradigms is inspiring. We would like to point that the main advantage of PaCE over the mentioned RLHF, SFT, and KE are two-fold.
>
>
>
> * Training-free: RLHF, SFT, KE all need to tune the parameters of LLM, which potentially degrade the well-structured priors of the pre-trained LLM. Taking a step back, even if LoRA is adopted for these paradigms, the training/tuning incur significant computation and memory costs. PaCE does not modify the parameters of LLM and requires no training. It better preserves priors of LLM, provides a low-resource alignment solution, and retains the general linguistic capabilities.
> * Interpretable and Adaptive: The solved coefficients are an accurate interpretation of how a user input’s representation is composed in the concept space. Also, when a new alignment goal is set, RLHF, SFT, and KE need to collect sufficient task samples and tune the LLM on the new dataset. In contrast, PaCE just needs to run the concept partitioner through PaCE-1M, which is expected to be much faster and more convenient.
>
>
>
>
>
> We are glad to respond to further comments or other concerns (if any) during the discussion period.  We appreciate your insightful suggestions which have helped to validate and strengthen our framework.
>
>
>
> Thanks and regards,
>
> Authors of Submission #8804
>
>
>
> [B1] _Representation Engineering: A Top-Down Approach to AI Transparency._ Zou et al., 2023.
>
> [B2] _Universal and Transferable Adversarial Attacks on Aligned Language Models._ Zou et al., 2023.
>
> [B3] _Interpretable by design: Learning predictors by composing interpretable queries_. Chattopadhyay, et al., 2022.
>
> [B4] _Leveraging sparse linear layers for debuggable deep networks._ Wong et al., 2021.

---

> ### Comment · Reviewer_s54k · 2024-08-13
> **Response**
>
> Thanks for the authors' response. Most of my concerns have been addressed. It would be great if the authors incorporate those insights into the revised version. I will increase the score from 5 to 6.

---

> > ### Author Response · Authors · 2024-08-13
> > **Thank you for your recognition and raising the score!**
> >
> > Dear Reviewer,
> >
> > We greatly appreciate your thoughtful feedback, your support of our work, and your decision to raise the score. We are especially grateful for your comments on the preparatory phase, valuable suggestions for an additional benchmark, and comprehensive insights on multiple alignment paradigms.
> >
> > As you suggested, we will revise manuscript $\S 4.1$ for AdvBench and adjust Appendix for (1) details of the preparatory phase, (2) dictionary input, and (3) connections among different paradigms. We will also revise other points as promised. Thank you again for your valuable input!
> >
> > Sincerely,
> >
> > Authors

---

### Official Review · Reviewer_Cdr5 · 2024-07-15

**Soundness:** 3
**Presentation:** 3
**Contribution:** 3
**Rating:** 6
**Confidence:** 4

**Summary:**

This paper presents a framework for aligning LLMs by using sparse coding techniques on a comprehensive concept dictionary. PaCE effectively controls and modifies neural activations of LLMs to achieve alignment goals such as response detoxification, faithfulness enhancement, and sentiment revising. The proposed method addresses limitations of existing alignment methods, such as costly fine-tuning, inadequate removal of undesirable concepts, and unnecessary removal of benign concepts. The paper demonstrates state-of-the-art performance in alignment tasks while maintaining the linguistic capabilities of LLMs.

**Strengths:**

Originality: The paper introduces a unique approach to LLM alignment by combining sparse coding with a large-scale concept dictionary. This method provides a novel perspective on activation engineering, differentiating itself from traditional parameter fine-tuning and prompt engineering approaches.
Quality: The methodology is well-developed, with thorough explanations and clear diagrams illustrating the concept construction, partitioning, and activation intervention processes. The use of sparse coding for accurate activation decomposition is well-motivated and justified.
Clarity: The paper is well-written, with a clear structure and logical flow. Each section builds on the previous one, providing a comprehensive understanding of the proposed framework. The experimental setup and results are presented in a detailed and easy-to-follow manner.
Significance: The proposed PaCE framework addresses critical issues in LLM alignment, offering a scalable and efficient solution. The state-of-the-art performance demonstrated in various alignment tasks highlights the practical significance and potential impact of this work on the broader AI community.

**Weaknesses:**

Computational Efficiency: While the paper demonstrates that PaCE outperforms existing methods in alignment tasks, the computational efficiency could be further optimized. The comparison with OrthoProj shows a significant improvement, but PaCE is still slower than VecAdd. Future work could focus on enhancing the speed of the proposed method.
Generalization to Other Models: The paper primarily evaluates PaCE on llama 2 7B, 13B models. It would be beneficial to demonstrate the generalizability of the framework across a wider range of LLMs to strengthen the claims of broad applicability.
Interpretability of Concept Dictionary: While the paper provides examples of concept vectors and their partitioning, a deeper analysis of the interpretability and semantic consistency of the concept dictionary would be valuable. Understanding the intrinsic properties of the concept vectors could offer further insights into the alignment process.
Reproducibility: The paper mentions the availability of the PaCE-1M dataset and plans to release the source code. Ensuring that all experimental details, including hyperparameters and specific configurations, are thoroughly documented will be crucial for reproducibility and wider adoption of the proposed framework.

**Questions:**

Computational Efficiency: Are there any ongoing efforts or planned approaches to optimize the computational efficiency of PaCE, particularly in comparison to VecAdd?
Model Generalization: Have you tested PaCE on other LLMs beyond llama 2 7B, 13B models? If so, could you provide insights or results from those experiments?
Concept Dictionary Analysis: Can you provide more detailed analyses or case studies on the interpretability and semantic consistency of the concept dictionary used in PaCE?
Ablation Studies: The ablation studies provide valuable insights into the contributions of various components of PaCE. Are there any additional ablation results, particularly focusing on the impact of different sparsity-promoting regularizers or dictionary sizes?

**Limitations:**

The authors adequately address the limitations and potential societal impacts of their work. They discuss the computational overhead and the need for extensive concept dictionaries, as well as the importance of ensuring ethical considerations in the alignment of LLMs. Constructive suggestions for improvement include exploring more efficient algorithms for sparse coding and investigating the scalability of PaCE to even larger concept dictionaries. The paper could also benefit from a deeper discussion on mitigating potential biases in the concept partitioning process.

---

> ### Author Rebuttal · Authors · 2024-08-06
>
> Dear Reviewer Cdr5,
>
> We greatly appreciate your constructive feedback and kind acknowledgment of the novelty and advantages of our approach. It is our pleasure to address your comments and provide clarification below.
>
> **Computational Efficiency**
>
> Thank you for acknowledging the task (e.g., detoxification) performance of PaCE. Yes, it is possible to make the method even faster, at the trade-off of task performance.  While our method jointly solves for the entire set of coefficients simultaneously, we can solve sparse decomposition in a greedy manner to speed up the computation. That is, the coefficients are obtained procedurally (i.e., one by one) in the order of energy minimization. The main benefit of using greedy algorithms is that the user can arbitrarily set the desired threshold on the number of non-zero coefficients in pursuit of better time efficiency. Based on such observation, we recap specific decomposition details in our paper and show an alternative speed-up version:
>
> * The elastic-net solver (Line 213) used in the experiments in $\S 4$ is designed to compute an exact solution of coefficients for the optimization problem (Equation (1), Line 186-189).  The coefficients are obtained simultaneously.
> * Motivated by your inquiry, we additionally evaluate Orthogonal Matching Pursuit (OMP), a fast greedy solver in the compressed sensing literature [A1, A2] for our activation decomposition. On a high level, OMP iteratively 1) adds to the support the concept that has maximum coherence with the unexplained residual, and 2) updates the residual by solving a least square using the new support. It stops when a pre-defined maximum size k of support is reached. Intuitively, the k is the number of non-zero elements in the solved coefficients.
>
> We compare two solvers on response detoxification ($\S 4.1$) in the table below.
>
> * Notably, with a small choice of k=50, OMP is an order of magnitude faster than Elastic Net, while it achieves a safety score 12.3% lower than that of Elastic Net.
> * Overall, the observations in this experiment validate that a greedy solver can improve computational speed at the cost of safety performance.
>
> |  | OMP (k=50) | OMP (k=100) | OMP (k=150) | OMP (k=200) | Elastic Net |
> |---|---|---|---|---|---|
> | Time per decomposition (s) | 0.045 | 0.182 | 0.381 | 0.749 | 0.411 |
> | Safety (↑) | 63.1 | 64.4 | 66.9 | 70.8 | 72.0 |
>
> **Generalization to Other LLMs**
>
> We evaluate PaCE on Mistral-7B-Instruct (version 1) for detoxification and observe PaCE’s superior performance while preserving linguistic capability. Mistral-7B’s strong instruction-following leads to a lower initial safety score and higher MMLU performance. Hence unlike LLaMA2-7B-Chat, the prompting baseline for detoxification does not significantly harm linguistic performance.
>
> |Mistral-7B-Instruct|Safety|Linguistic|Capability||
> |:---:|:---:|:---:|:---:|:---:|
> |Method / Metrics|Average (%, ↑)|Fluency (↑)| Perplexity (↓) | MMLU (%, ↑)  |
> |Vanilla | 5.20 | 6.91 | 3.57 | 56.4 |
> |Prompting | 54.8 | 6.80 | 3.58 | 54.4 |
> |VecAdd | 64.7 | 6.69 | 4.23 | 44.3 |
> |OrthoProj | 65.2 | 6.74 | 4.35 | 44.9 |
> |PaCE (Ours) | 76.3 | 6.89 | 4.19 | 46.1 |
>
> **Interpretability of Concept Dictionary**
>
> Figure 8 and Appendix D.2 show semantic similarity between concept vectors by clustering the first 10,000 concepts in our PaCE-1M dictionary. The observed semantic structures (salient clusters and decision boundaries) indicate that the target LLM has an activation space that understands and organizes semantic information of the concepts, enabling further analysis and manipulations in PaCE. Figure 10 and Appendix D.3 show the activation space's utility for concept retrieval, indicating close coherence between the target and relevant concept vectors. With full respect to your and Reviewer s54k’s insights on this, we will add a new section Appendix D.4 for discussions of analyzing other properties (e.g., sparsity, magnitude, direction).
>
>
>
> **Ablation Study**
>
> Thank you for inquiring about ablation studies on key components of PaCE. Figure 6 shows detoxification performance for LLaMA2-13B-Chat across different dictionary sizes, with safety scores increasing and converging around 9000-10000. For the regularization design, we evaluated different setups of $\tau$ in Equation (2) that implement the sparsity-prompting regularizer. The results are shown in the table below:
>
>
> | $\tau$ | 0 | 0.35 | 0.65 | 0.95 | 1.0 |
> |---|---|---|---|---|---|
> | Note | Pure $\ell_2$ | N.A. | N.A.   | N.A.      | Pure $\ell_1$ |
> | Safety (↑)     | 68.9 | 65.4 | 71.6 | 72.0 | 66.5 |
>
>
> The results show that the regularization with $\tau$=0.95 (our choice stated in the paper Appendix B.4) yields the best safety performance among the five choices. Pure ridge regression ($\tau$=0) and pure lasso regression ($\tau$=1) do not perform as well as the mixed regularization strategy.
>
>
> **Reproducibility**
>
> We appreciate your emphasis on reproducibility. Appendix B.4 details our frameworks, experiments, and hyper-parameters for inferring on open-source LLMs, concept curation, and knowledge retrieval. We have also included details of our computing resources on GPUs. Upon acceptance, we will open-source the PaCE-1M dataset and PaCE implementation with documentation. We will continue maintaining the project (e.g., computing concept dictionaries and partitioning the concepts with new expert models) for future open-source LLMs and new alignment tasks.
>
>
> We will include the experiments above in $\S4$ and Appendix B.5 of the paper, and we will revise the writing based on your suggestions. We appreciate your valuable insights which have helped to validate and strengthen our framework.
>
> Thanks and regards,
>
> Authors of Submission #8804
>
> [A1]  _Orthogonal matching pursuit: Recursive function approximation with applications to wavelet decomposition._ Pati et al., 1993.
>
> [A2] _Orthogonal matching pursuit for sparse signal recovery with noise._ Cai et al., 2011.

---

> > ### Author Response · Authors · 2024-08-13
> >
> > Dear Reviewer Cdr5,
> >
> > Thanks again for your time and efforts in writing the review. We addressed your concerns in detail in the rebuttal, and we hope the response covers the comments. We are more than happy to further clarify or address additional questions. Please let us know if you still have any unclear parts of our work.
> >
> > Sincerely,
> >
> > Authors

---

### Author Rebuttal · Authors · 2024-08-07

### Global Response

We thank all reviewers for their valuable time and intellectual input on the paper. We take the opportunity to provide a global summary here and communicate with the reviewers in the individual rebuttals.



**Presentation Quality**

As per reviewers’ comments, the paper has _thorough explanations and clear diagrams_ (Cdr5), provides _clear visualization and detailed interpretation to illustrate its underlying principles_ (75Pi), gives _comprehensive and clear articulations_ (rWAy), and is _well-written with a clear structure and logical flow_ (Cdr5).

We also appreciate Reviewer s54k’s contributive inquiry about additional details of activation alignment, and Reviewer rWAy’s comments on baseline implementation. We have addressed them in the individual rebuttals.



**Novelty and Significance**

Our Parsimonious Concept Engineering (PaCE) sparsely decomposes a target LLM representation on a large-scale concept dictionary to precisely re-orient the LLM behavior and effectively improve its trustworthiness.



* New dataset. We collect a large-scale concept representation dataset, PaCE-1M, that consists of 40,000 concepts extracted from over 1,200,000 context sentences. It is generalizable to multiple LLMs and its concepts are annotated for downstream tasks such as detoxification and sentiment revising.
* Novel method. We decompose the neural activations as a sparse linear combination of these concept directions using efficient sparse coding techniques. The decomposition provides an effective and accurate estimate of both undesirable and benign components in the target representation, which is often overlooked in previous activation engineering methods.

Indeed, the novelty of the proposed framework and the innovations in the pipeline are widely recognized by multiple reviewers. We appreciate the reviewers' comments that the proposed PaCE framework is _novel_ (s54k, rWAy, Cdr5), _different from the existing methods_ (rWAy), _unique, well-motivated, and well-developed_ (Cdr5), _adaptive and insightful_ (s54k).

In the individual rebuttals, we address valuable comments for concept polysemy and questions on framework design for activation manipulation (75Pi), and we elaborate on design choices of sparse optimization (Cdr5). We also respond to the insightful observations on the position of PaCE among secure alignment schemes (s54k).


**Experiments**

To validate our proposed framework, we evaluate PaCE on multiple alignment tasks including response detoxification, faithfulness enhancement, and sentiment revising ($\S4.1$, $\S4.2$). We show that PaCE achieves state-of-the-art performance on these tasks while retaining its linguistic capability at a comparable level. We further investigate the LLM activation space by PaCE-1M samples, showing the geometric consistency of concept semantics and the interpretability of the PaCE decomposition ($\S4.3$).

It is encouraging to have reviewers’ comments that the proposed approach is _showing improved alignment performance_ and _achieving outstanding results with 13B model_ (75Pi), and the framework is _state-of-the-art, scalable_ and _efficient_ (Cdr5). They also find that the _evaluation is comprehensive_ and _the results are insightful_ (s54k), and the _superiority of the proposed method in pursuing alignment goals_ is validated (rWAy).

We are grateful to receive reviewers’ suggestions about computational efficiency (Cdr5, rWAy), additional malicious jailbreaks (s54k), additional benign (helpfulness) benchmarks (75Pi), and other target LLMs (Cdr5, 75Pi). These inquiries on experiments help us to further confirm the general applicability and outperformance of our framework, and we have addressed all of them in the individual rebuttals.

 **Summary**

In each of the individual rebuttals below, we address the reviewers’ valuable suggestions on paper presentation, framework validity, additional experiments, and all other perspectives. All insights and questions are highly important to the continual improvement of our work, and based on them we have made or will make multiple revisions to the manuscript and appendix as promised.



In summary, we are more than happy to receive four high-quality, solid, and insightful reviews this time; thank you all. Please feel free to communicate with us during the discussion period if you have any further questions.

---

### Decision · Program_Chairs · 2024-09-25

**Decision:**

Accept (poster)

**Comment:**

The paper presents a nice contribution in the area of language model editing. I believe the idea itself is solid, even though the core idea has many instances in the literature (editing activations through projection). Most of the reviewers were happy with the paper overall, and the authors clearly put a lot of work into improving it further.